# Structures and gating mechanisms of human bestrophin anion channels

Aaron P. Owji[1,2,6], Jiali Wang [1,6], Alec Kittredge [1,2], Zada Clark[1], Yu Zhang [1✉], Wayne A. Hendrickson [3,4,5✉] & Tingting Yang [1✉]

Bestrophin-1 (Best1) and bestrophin-2 (Best2) are two members of the bestrophin family of calcium ($Ca^{2+}$)-activated chloride ($Cl^-$) channels with critical involvement in ocular physiology and direct pathological relevance. Here, we report cryo-EM structures of wild-type human Best1 and Best2 in various states at up to 1.8 Å resolution. $Ca^{2+}$-bound Best1 structures illustrate partially open conformations at the two $Ca^{2+}$-dependent gates of the channels, in contrast to the fully open conformations observed in $Ca^{2+}$-bound Best2, which is in accord with the significantly smaller currents conducted by Best1 in electrophysiological recordings. Comparison of the closed and open states reveals a C-terminal auto-inhibitory segment (AS), which constricts the channel concentrically by wrapping around the channel periphery in an inter-protomer manner and must be released to allow channel opening. Our results demonstrate that removing the AS from Best1 and Best2 results in truncation mutants with similar activities, while swapping the AS between Best1 and Best2 results in chimeric mutants with swapped activities, underlying a key role of the AS in determining paralog specificity among bestrophins.

[1] Department of Ophthalmology, Columbia University, New York, NY, USA. [2] Department of Pharmacology, Columbia University, New York, NY, USA. [3] Department of Biochemistry and Molecular Biophysics, Columbia University, New York, NY, USA. [4] Department of Physiology and Cellular Biophysics, Columbia University, New York, NY, USA. [5] New York Structural Biology Center, New York, NY, USA. [6] These authors contributed equally: Aaron P. Owji, Jiali Wang. ✉email: yz3802@cumc.columbia.edu; wah2@cumc.columbia.edu; ty2190@cumc.columbia.edu

The bestrophins are a family of $Ca^{2+}$-activated anion channels consisting of four members (Best1–4) in mammals[1]. They are widely distributed in various human organs including the airways, colon, kidney, pancreas and central nervous system, but best known for their physiological roles in the eye. In particular, Best1 is predominantly expressed in retinal pigment epithelium (RPE) and genetically linked to a spectrum of retinal degenerative disorders collectively known as bestrophinopathies[2–4]. To date, over 250 different mutations in Best1 have been identified to cause bestrophinopathies, and the patients are susceptible to progressive vision loss that may eventually lead to blindness[2,5–7]. On the other hand, Best2 resides in non-pigmented epithelium (NPE) regulating intra-ocular pressure (IOP), which is significantly reduced in $Best2^{-/-}$ mice[8–10]. As elevated IOP is a deleterious condition and a major risk factor for glaucoma, Best2 represents a potential pharmaceutical target. Therefore, understanding the structure and function of bestrophin channels holds tremendous biomedical significance.

Functionally, distinct channel activities have been observed among bestrophin paralogs from the same species[11,12], as exemplified by the significantly smaller currents conducted by human Best1 (hBest1) in transiently transfected HEK293 cells compared to those from human Best2 (hBest2) under the same conditions[12], but what determines paralog specificity is unknown.

The structures of a bacterial bestrophin homolog from *Klebsiella pneumoniae* (KpBest), a Best1 homolog from chicken (cBest1), and a Best2 homolog from bovine (bBest2) have been reported[12–14], but no human or any wild-type (WT) bestrophin structure is available. All three of them are homopentamers and exhibit a flower vase-shaped ion conducting pathway with two major permeation constrictions, the "neck" at the transmembrane pore and the "aperture" at the cytosolic exit[12–14]. The neck is formed by three highly conserved hydrophobic residues from each of the five protomers, corresponding to I62/I66/F70 in KpBest and I76/F80/F84 in cBest1 and bBest2[12–14]. By contrast, the aperture displays significant divergence among species and paralogs, as it is formed by I180 in KpBest, V205 in cBest1, and a K208-E212 pair in bBest2[12–14]. Therefore, although a conserved overall architecture of bestrophin channels has been documented by previously solved homolog structures, the specific compositions of some key elements in human bestrophins, notably the aperture, which would likely inform functional specificity, remain unclear.

Moreover, bestrophins are activated and inactivated in response to low and high $Ca^{2+}$, respectively[11,15,16], but the mechanism of $Ca^{2+}$-dependent channel gating remains largely elusive. Ample electrophysiological evidence indicates that the neck and aperture serve as dual $Ca^{2+}$-dependent channel "gates", as disrupting either of them partially impairs $Ca^{2+}$-dependence while disrupting both makes the channel completely $Ca^{2+}$-independent[12,14,17,18]. Hence, the mechanisms of bestrophin channel activation and inactivation ultimately come down to how the neck and aperture transition from closed to open, and vice versa. However, the structural bases of these processes are still poorly understood, leading to several outstanding questions in the field. Firstly, the aperture stays in the closed conformation in all previous bestrophin structures regardless of the presence of $Ca^{2+}$, leaving the open conformation of this channel gate yet to be discovered[12–14,19]. Secondly, a conserved XXSFXGS auto-inhibitory motif in the C-terminal cytosolic region has been identified to play a key role in regulating both activation and inactivation of bestrophins, as mutants omitting this motif conduct significantly elevated currents with diminished rundown[11,20], but the mechanism of how this motif prevents the channel from opening is poorly characterized. Thirdly, the only

open conformation of the neck is observed with two C-terminally truncated cBest1 derivatives, cBest1$_{1–345}$ and cBest1$_{1–345}$-W287F[19], leaving in question whether the neck opens in the same manner in the WT bestrophins.

In this study, we solved the high-resolution cryo-EM structures of WT hBest1 and hBest2 in the $Ca^{2+}$-unbound closed, $Ca^{2+}$-bound closed, $Ca^{2+}$-bound partially open and fully open states. These structures illustrate the unique composition of the human bestrophin apertures, which are formed by I205/Q208/N212 in hBest1 and S205/K208/E212 in hBest2, and partially open conformations at both gates of hBest1 compared to fully open gates in hBest2. Importantly, a crucial 34-residue auto-inhibitory segment (AS) containing the previously identified auto-inhibitory motif was defined, which provides an additional layer of regulation for gating of the neck besides the $Ca^{2+}$-sensor. Structure-inspired electrophysiological experiments confirm the functional significance of the aperture-forming residues and the AS, and strongly support a model in which residues 1–345 constitute the conserved core unit of bestrophin channels while the AS (residues 346–379) determines the functional specificity among species/paralogs.

## Results

### High-resolution cryo-EM structures of human Best1 and Best2 (hBest1/hBest2).
To elucidate the architecture of human bestrophins, we purified WT hBest1 and hBest2 for single-particle cryo-EM analysis in the absence of $Ca^{2+}$ (EGTA), or after a 30 s spike of 1 μM or 5 mM $Ca^{2+}$. The structures were solved at 1.8–2.3 Å resolution except for hBest1 with 20 mM EGTA at 3.1 Å (Fig. 1a, b, Supplementary Figs. 1, 2, and Supplementary Table 1). Remarkably, although the overall hBest1 and hBest2-structures resemble the pentameric assembly shown by previously solved bestrophins[12–14], different conformations are observed within subsets of $Ca^{2+}$-bound particles at the two landmark permeation constrictions in the ion conducting pathway, the neck and aperture (Fig. 1c–f), revealing the gating mechanisms of bestrophin channels.

### Partially and fully open states of the neck.
In both hBest1 and hBest2 structures, the neck is formed by the same set of hydrophobic residues (I76, F80 and F84) as those previously reported in bBest2 and cBest1[12,13], underlining a highly conserved neck in bestrophins (Fig. 1c–f). Under both 1 μM and 5 mM $Ca^{2+}$ conditions, most of the hBest1/hBest2 particles (~90–95%) were in a $Ca^{2+}$-bound closed state (Table 1, Figs. 1c, d, 2a), which is very similar to the $Ca^{2+}$-unbound closed state. Strikingly, however, a small population of hBest1 and hBest2 particles exhibited a partially and fully open neck (Figs. 1e–h, 2b, c, and Supplementary Figs. 1, 2), respectively.

A fully dilated neck is present in 6.3% and 12% of hBest2 particles exposed to 1 μM and 5 mM $Ca^{2+}$ spike, respectively (Supplementary Fig. 2). This conformation resembles that of the previously solved cBest1$_{1–345}$ and cBest1$_{1–345}$-W287F[19]. Critical conformational changes take place at residues Y236, F282, F283, and W287, concomitant with the neck residues I76, F80 and F84 turning away from the ion conducting pore, drastically dilating the minimum radius of the neck from 0.8 Å to 4.5 Å and revealing a hydrophilic ion permeation pathway through the membrane (Figs. 1f, h, 2c). By contrast, a partially open neck is present in 5.7% and 10.9% of hBest1 particles exposed to 1 μM and 5 mM $Ca^{2+}$ spike, respectively (Supplementary Fig. 1), representing an intermediate open state. In this intermediate conformation, the same structural changes take place at residues F282 and F283 as in the fully open state. This allows neck residue I76 to flip away from the central axis (radius from 0.8 Å to > 6.0 Å), while Y236

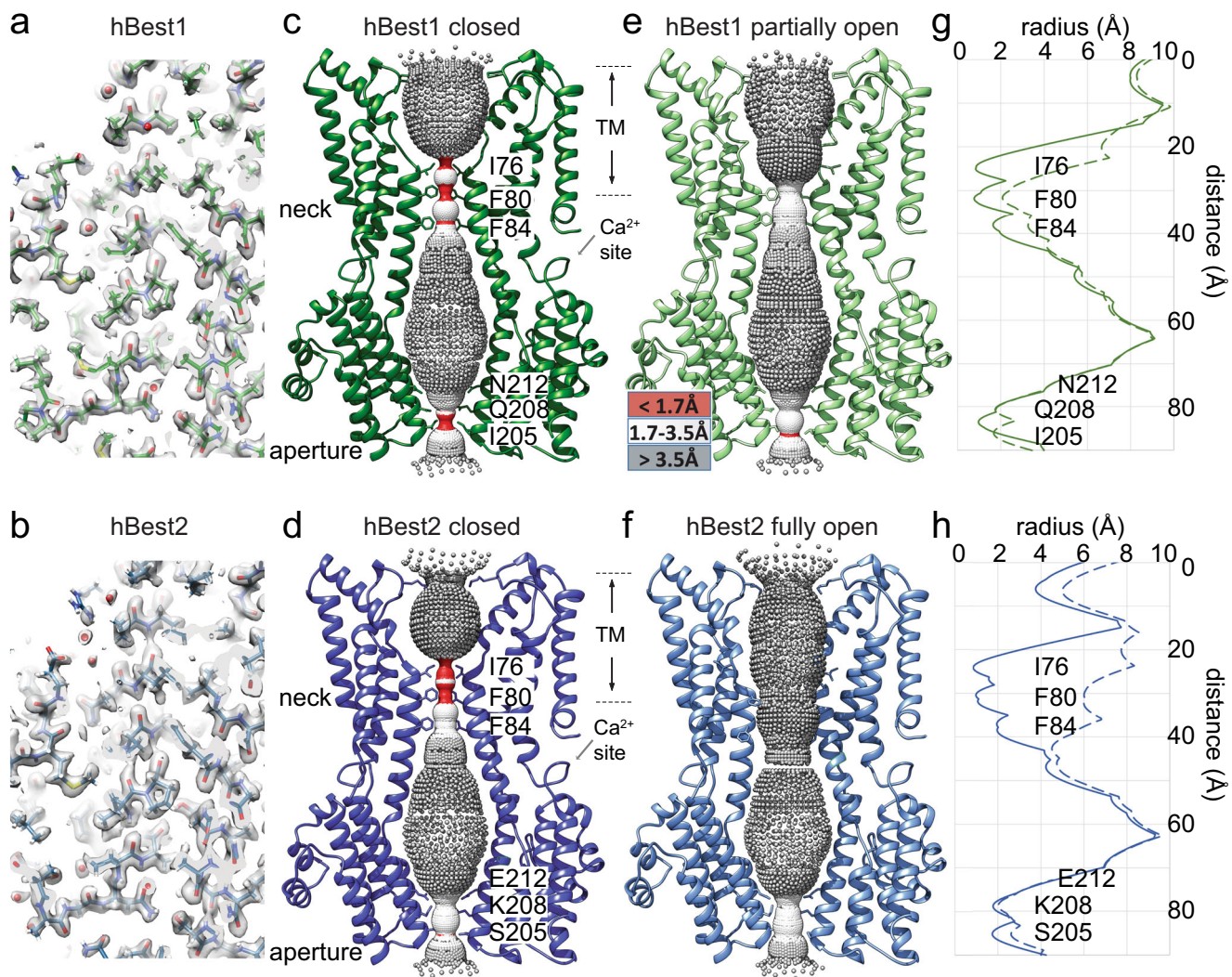

**Fig. 1 Cryo-EM structures of hBest1 and hBest2 in different states. a, b** Example densities from the Ca²⁺-bound (1 μM) maps of hBest1 **a** and hBest2 **b** exhibiting high resolution features, including carbonyl groups, water molecules (red spheres), and holes in prolines. **c–f** Side view of two opposing (144°) protomers from hBest1 in the closed **c** or partially open **e** states and hBest2 in the closed **d** or fully open **f** states, with the ion permeation pathway visualized and pore diameter depicted as colored dots: the tighter the pore radius, the smaller and denser the dots. *Inset*, color key. Major constrictions to the ion permeation pathway are shown as sticks with residue labels on the right. Dashed lines indicate approximate boundaries of the transmembrane domains. **g, h** HOLE graph of the ion permeation pathway radius as a function of distance along the channel axis of hBest1 **g** and hBest2 **h** in the Ca²⁺-bound closed (solid lines) and partially/fully open (dashed lines) states. HOLE and Chimera are used to compute pore dimensions and visualize pore densities, respectively.

| Table 1 Composition of bestrophin particles under different Ca²⁺ conditions. | | | | |
|---|---|---|---|---|
| | **1 μM Ca²⁺** | | **5 mM Ca²⁺** | |
| | Closed (%) | Open (%) | Closed (%) | Open (%) |
| **hBest1** | 94.3 | 5.7 (partially) | 89.1 | 10.9 (partially) |
| **hBest2** | 93.7 | 6.3 (fully) | 88.0 | 12.0 (fully) |

and W287 only undertake a small conformational change with intermediate dilation of F80 (from 0.7 Å to 2.1 Å) and F84 (from 1.6 Å to 3.2 Å) as the helix unwinds at P77. Consistently, an ion-like density, which is absent in the closed hBest1 structures or the fully open hBest2 and cBest1₁₋₃₄₅ structures, is present at the center of F84, potentially representing a dehydrated Cl⁻ ion (Fig. 2b and Supplementary Fig. 1).

**Structural analysis of the C-terminal segment.** A crucial contribution of the C-terminal segment to the gating of bestrophins has been well established in previous studies[11,15,20], but the underlying mechanism and structural basis remain unclear, as the only bestrophin structures with an open neck were obtained from C-terminally truncated cBest1 containing just the first 345 residues of the channel[19]. By comparing Ca²⁺-bound hBest2 in the closed and open states, we defined the structural changes coordinated with neck gating at residues 346–379. In the closed state, this region is well-ordered and wraps around the circumference of the cytosolic domain, extending clockwise around the channel when viewed from the membrane (Fig. 3a). There are three major contact points on the channel periphery: the backbone carbonyl oxygen of S363 of protomer −1 directly contacts the side chain of A7 of protomer 0, pushing the Ca²⁺-clasp, which must move ~0.5 Å root-mean-square deviation (RMSD) away from the central axis to allow neck dilation, towards the central axis; the side

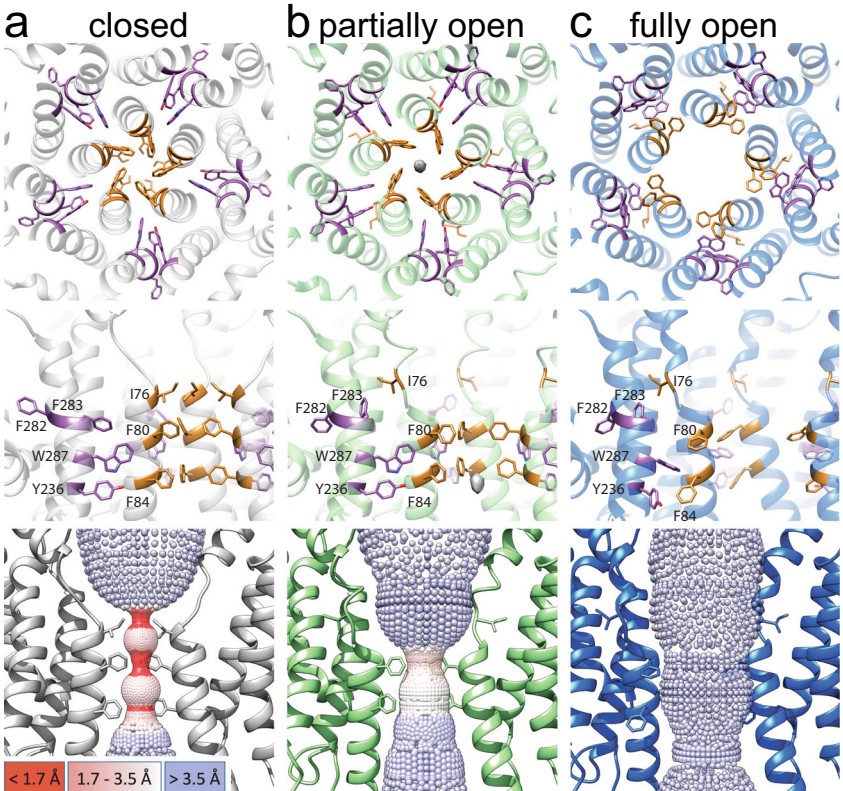

**Fig. 2 Different conformations of the neck. a** Molecular model of the bottom (top row) and side (middle row) views of the neck in the Ca$^{2+}$-bound closed state (hBest1). The neck residues (I76, F80, F84) and critical residues involved in neck gating (Y236, F282, F283, W287) are labeled. Bottom row, closed state model with pore diameter depicted as colored dots. Tighter pore radius has smaller, denser, red dots, while a less restrictive radius has larger, less dense, blue dots. *Inset*, color key. **b** Models for the Ca$^{2+}$-bound partially open state (hBest1), in the same format as in **a**. An ion-like density is captured at F84 of the partially open neck in hBest1. Density shown from C1 refinement. **c** Models for the Ca$^{2+}$-bound fully open state (hBest2), in the same format as in **a**.

chain of F376 of protomer −1 is wedged into a hydrophobic pocket formed by H156, A160 and F151 of protomer +1, and P346 and T348 of protomer 0; a salt bridge is formed between Q357 of protomer −1 and E306 of protomer 0 (Fig. 3b–d and Supplementary Fig. 3). Therefore, the binding of residues 346–379 to the channel periphery provides an inter-protomer cooperative mechanism to constrict the channel concentrically. In the open state, by sharp contrast, residues 346–356 (Fig. 3, yellow) and 369–379 (Fig. 3, red) become fully disordered while moderate density remains for anchor residues 357–368 (Fig. 3, blue), indicating diminished binding of this region to the channel periphery. Consistently, in 3D variability analysis (3DVA) with the Ca$^{2+}$-bound hBest2 datasets, neck dilation occurs simultaneously with synchronized disordering of residues 346–356 (yellow) and 369–379 (red), while residues 357–368 (blue) remain partially ordered and partially bound to the channel (Fig. 3e, f and Supplementary Movie 1). These results suggest that residues 357–368 (blue) act as an anchor responsible for holding the C-terminal segment to the channel core, while the flanking residues 346–356 and 369–379 cooperatively stay bound and unbound to regulate channel gating in a reversible manner. Importantly, residues 369–379 of protomer 0 binds to residues 346–356 of the +1 protomer, which is proximal to the Ca$^{2+}$ clasp of the +2 protomer (Fig. 3a–c), establishing a concentric constriction of the neck which must be unleashed to allow channel opening.

Consistent results were obtained from structural analysis of hBest1 Ca$^{2+}$-bound datasets, except that residues 346–355 (yellow) and 368–378 (red) were relatively less disordered

compared to their counterparts in hBest2 (Fig. 3g, h and Supplementary Movie 2), suggesting a stronger binding of this region to the channel periphery in hBest1. This is in accord with the partially and fully open neck in hBest1 and hBest2, respectively. Notably, a direct contact between F354 and E300 in hBest1 is disrupted in hBest2 due to an insertion of F at position 353, which also offsets the residue numbers of hBest1 and hBest2 by one within this region (Supplementary Fig. 3).

As the critical role of residues 346–378/379 in channel gating is conserved in hBest1 and hBest2, and the degree of their displacement from the channel periphery correlates with the level of neck dilation in the presence of Ca$^{2+}$, we termed this region the "auto-inhibitory segment" (AS). It consists of an "Anchor" (residues 356–367 in hBest1 and 357–368 in hBest2, blue), which remains at least partially bound to the channel periphery, and two flanking "AS Cooperativity Regions" (ACR1, residues 346–355 in hBest1 and 346–356 in hBest2, yellow; ACR2, residues 368–378 in hBest1 369–379 in hBest2, red), which undergo synchronized dissociation from the channel periphery to allow neck dilation (Fig. 3a, e–h). 3DVA of the Ca$^{2+}$-unbound datasets revealed that the AS undergoes the same ordering/disordering motion as observed in the Ca$^{2+}$-bound datasets, although there is no opening of the neck, suggesting that the binding of the AS to the channel periphery does not require Ca$^{2+}$, and unleashing the constriction imposed by the AS is necessary but not sufficient for neck opening. Taken together, our results established the role of the AS as an essential gating element that works together with the Ca$^{2+}$-sensor to control closing/opening of the neck.

**Fig. 3 The AS constricts the channel concentrically. a**, **b** Color-coded schematic showing the specific segments of the AS and the channel core of protomer 0 colored copper. Protomers are numbered. Green dots, $Ca^{2+}$ ions. **c** Closeup of ACR2 of protomer −1 interacting with ACR1 of protomer 0 proximal to the $Ca^{2+}$-clasp of protomer +1. **d** Conformational change of F84 illustrated by superimposition of the closed and open states of $Ca^{2+}$-bound hBest2. Residues of the AS contributing to neck closure by constriction of the $Ca^{2+}$ clasp are shown. **e–h** Coordinated ordering and disordering of the AS between the $Ca^{2+}$-bound closed and open states of hBest2 from side **e** and top **f** views and hBest1 from side **g** and top **h** views. Residues of the neck are colored pink.

**Bestrophin structures in the absence of the AS.** To further investigate the structural significance of the AS, we generated hBest1$_{1-345}$ and bBest2$_{1-345}$, and solved their structures in the presence of 5 mM $Ca^{2+}$ at 2.4 Å and 1.9 Å, respectively. Remarkably, both structures exhibited 100% of particles with a fully open neck (Fig. 4 and Supplementary Fig. 4), in sharp contrast to the only ~10% particles with partially or fully open neck in WT hBest1 and hBest2, respectively. No conformational variability was detected by 3DVA, suggesting a highly stable open conformation in these structures. These results reaffirm an essential role of the AS in constraining the neck, and demonstrate the existence of the fully open conformation of the hBest1 neck upon absence of the AS.

**Functional analysis of the C-terminal segment.** To test the functional significance of these C-terminal components, we truncated hBest1 at positions 345, 355, 367, 378 and 405, and

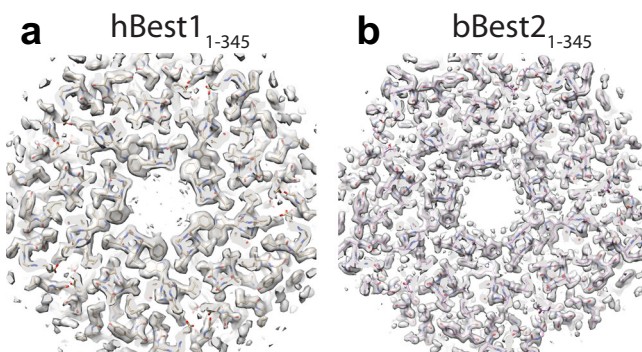

**Fig. 4 Structures of AS truncated mutants. a**, **b** Top view of cryo-EM map and model showing fully open necks of hBest1$_{1-345}$ **a** and bBest2$_{1-345}$ **b**.

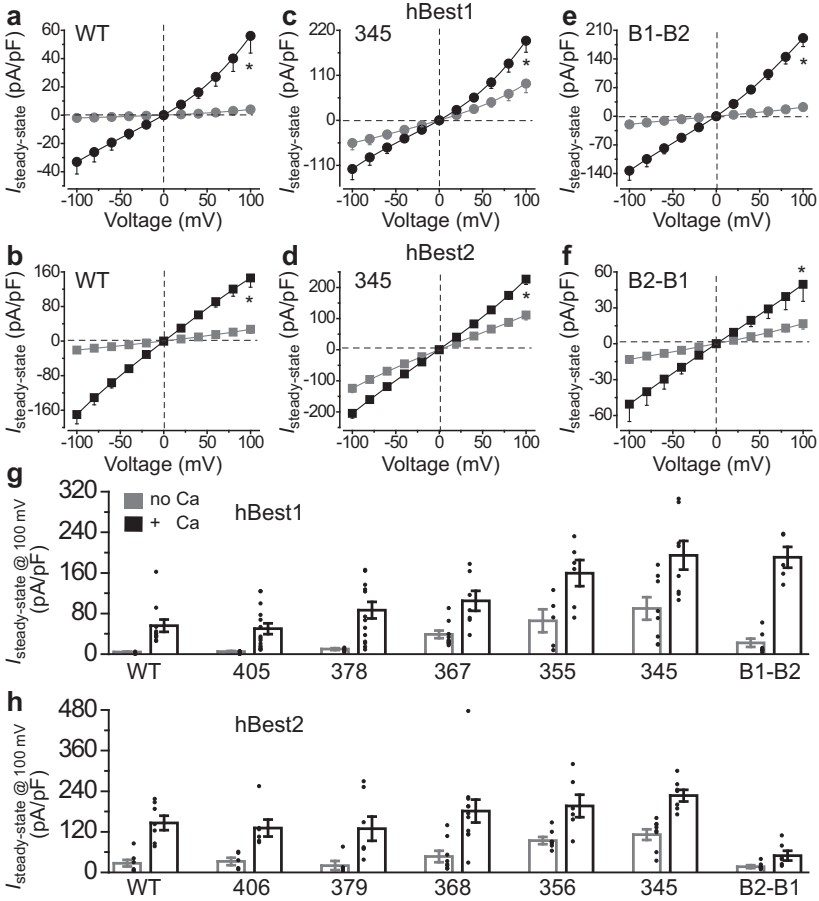

**Fig. 5 Ca$^{2+}$-dependent Cl$^-$ currents mediated by hBest1 and hBest2. a–f** Population steady-state current density-voltage relationships in the absence (gray) and presence (black) of 1 µM Ca$^{2+}$ from HEK293 cells expressing WT hBest1 **a**, WT hBest2 **b**, hBest1$_{1-345}$ **c**, hBest2$_{1-345}$ **d**, hBest1$_{1-345}$-hBest2$_{346-406}$ **e**, or hBest2$_{1-345}$-hBest1$_{346-405}$ **f**; $n = 5$–11 biologically independent cells for each point. *$P < 0.05$ compared to currents without Ca$^{2+}$, using two-tailed unpaired Student $t$ test. **g, h** Bar chart showing the steady-state current densities from HEK293 cells expressing the indicated hBest1 **g** or hBest2 **h** channels in the absence and presence of 1 µM Ca$^{2+}$, $n = 5$–15 biologically independent cells for each bar. B1-B2: hBest1$_{1-345}$-hBest2$_{346-406}$, B2-B1: hBest2$_{1-345}$-hBest1$_{346-405}$. Data are presented as mean values ± SEM. Source data and the precise n and $P$ values are provided in the Source Data file.

hBest2 at positions 345, 356, 368, 379 and 406 for electrophysiological analysis. Previously, we found that bBest2$_{1-406}$ conducts similar Cl$^-$ currents to those from the full-length bBest2[12], suggesting an insignificant role of the residues after position 406 in channel gating.

The full-length (FL) and truncated channels were individually transfected into HEK293 cells and subjected to whole-cell patch clamp recording in the absence and presence of 1 µM Ca$^{2+}$. As previously reported, hBest2-FL displayed significantly bigger currents than hBest1-FL under both conditions, while these two proteins had similar overall and membrane expression levels (Fig. 5a, b and Supplementary Figs. 5a, b and 6a, b)[12]. This is in accord with the fully vs. partially open neck in their structures in the presence of Ca$^{2+}$ and the more stably bound AS in hBest1 (Figs. 2b, c, 3e–h, and Supplementary Fig. 7). Remarkably, hBest1$_{1-345}$ and hBest2$_{1-345}$ were indistinguishable, both displaying robust leak currents with similar amplitudes in the absence of Ca$^{2+}$ and significantly increased currents with similar amplitudes at 1 µM Ca$^{2+}$ compared to the respective FL channels (Fig. 5c, d, g, h and Supplementary Fig. 6c–e, using two-tailed unpaired Student $t$ test), which is consistent with the 100% fully open neck in their structures. By contrast, currents from hBest1$_{1-378}$/hBest1$_{1-405}$ and hBest2$_{1-379}$/hBest2$_{1-406}$ are similar to those from hBest1-FL and hBest2-FL, respectively (Supplementary Fig. 8a, b,

g, h), reaffirming a critical role of the AS in regulating bestrophin function.

The currents from hBest1$_{1-355}$ resembled those from hBest1$_{1-345}$ (Fig. 5g, h and Supplementary Fig. 8e, g, h), suggesting that ACR1 is non-functional by itself without the other two components of the AS. The currents from hBest1$_{1-367}$ were intermediate between those from hBest1-FL and hBest1$_{1-345}$ (Fig. 5g, h and Supplementary Fig. 8c, g, h), suggesting that the combination of ACR1 + Anchor is partially functional without ACR2. Consistently, hBest2$_{1-356}$ and hBest2$_{1-368}$ both showed an intermediate current amplitude between hBest2-FL and hBest2$_{1-345}$ (Fig. 5g, h and Supplementary Fig. 8d, f–h), although not significantly different compared to either. This is likely due to the less stably bound AS and higher basal current amplitude in hBest2-FL than in hBest1-FL.

Furthermore, we swapped the 346–405/406 regions of the two channels, generating hBest1$_{1-345}$-hBest2$_{346-406}$ and hBest2$_{1-345}$-hBest1$_{346-405}$. In transiently transfected HEK293 cells, the whole cell currents of each chimeric channel resembled that of the corresponding channel from which the C-terminus originated (Fig. 5e–h and Supplementary Fig. 8g, h). Notably, the membrane expression levels of the truncation and swapping mutants were not significantly different when compared between each other or to the FL channels (Supplementary Fig. 5a, b). These results

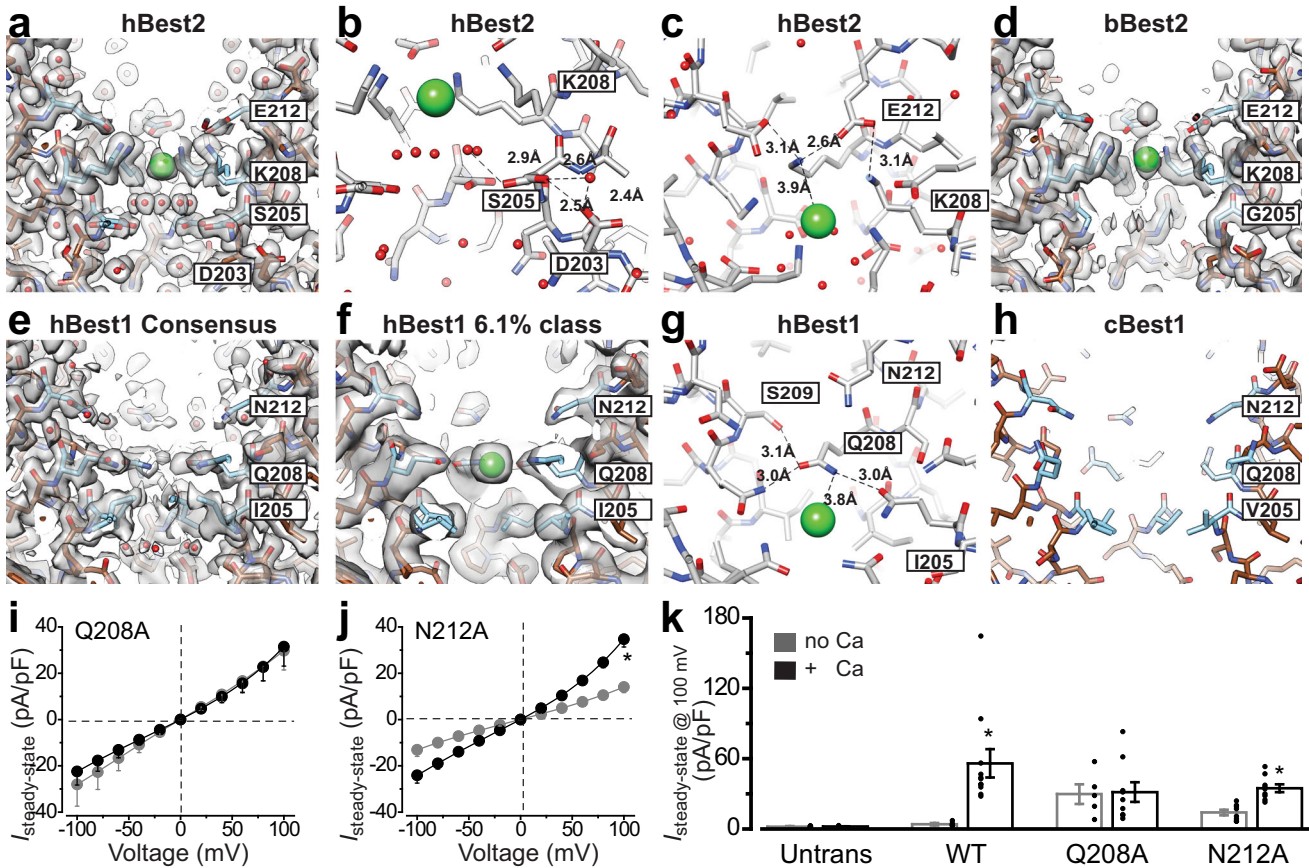

**Fig. 6 The aperture in different bestrophin structures.** Comparison of the aperture in Best2 **a–d** and Best1 **e–h** structures. The green dot represents a $Cl^-$ ion. **a** Side view of cryo-EM density map and model for hBest2. **b** hBest2 aperture viewed perpendicular to the central axis with dual conformations of S205. Bond distances between S205, D203, and water molecules are shown. **c** hBest2 bond distances between E212, K208, and a putative $Cl^-$ ion. **d** Cryo-EM density map and model for bBest2 side view. **e, f** Side view of cryo-EM density map and model for hBest1 consensus structure **e**, and a rare class with a putative $Cl^-$ ion bound at Q208 **f**. **g** Bond distances are shown for Q208, S209, and a putative $Cl^-$ ion in the same view as **c**. **h** Side view of cBest1 X-ray structure (4RDQ). **i, j** Population steady-state current density-voltage relationships in the absence (gray) and presence (black) of 1 μM $Ca^{2+}$ from HEK293 cells expressing hBest1 Q208A **i** and N212A **j**; $n = 5–9$ biologically independent cells for each point. *$P < 0.05$ compared to currents without $Ca^{2+}$, using two-tailed unpaired Student $t$ test. **k** Bar chart showing the steady-state current densities from HEK293 cells expressing the indicated channels in the absence and presence of 1 μM $Ca^{2+}$, $n = 5–11$ biologically independent cells for each bar. *$P < 0.05$ compared to currents without $Ca^{2+}$, using two-tailed unpaired Student $t$ test. Data are presented as mean values ± SEM. Source data and the precise n and $P$ values are provided in the Source Data file.

strongly suggest that the C-terminal AS is a key determinant of bestrophin paralog specificity.

**Divergent composition and conformation of the aperture.** In sharp contrast to the highly conserved composition of the neck, the aperture exhibits divergence among bestrophin paralogs, as it is formed by V205 in cBest1 and K208-E212 in bBest2[12,13].

The hBest2 aperture recapitulates structural features previously seen in bBest2: the K208-E212 pair, connected by a salt bridge, forms a constriction in the ion conduction pathway within the cytosolic region (Fig. 6a–d); an ion-like density is captured at the level of K208 (Fig. 6a–d and Supplementary Fig. 9a), possibly representing a passing $Cl^-$. The high similarity between hBest2 and bBest2 structures suggests a critical involvement of K208-E212 in hBest2 channel gating and selectivity, as we previously reported for bBest2 K208-E212[12]. Interestingly, S205 of hBest2 adopts two conformations in the $Ca^{2+}$-bound state, such that it can point directly into or away from the central axis (Fig. 6a, b), imposing a constriction with radius of 1.8 Å and 2.6 Å, respectively. As S205 is conserved in most mammalian Best2 homologs and Best2 is reported to conduct $HCO_3^-$ [21], the dual conformation of S205 may affect the flow of larger anions.

On the other hand, the hBest1 aperture exhibits several distinctive features compared to cBest1. Firstly, I205 is the narrowest point (radius 0.9 Å) within the cytosolic region of the ion conducting pathway, similar to I180 in KpBest, whereas the counterpart in cBest1 is V205 (radius 1.3 Å, Supplementary Fig. 9)[13]. Secondly, the Q208-N212 pair in hBest1 is connected by a water network in a similar manner to the salt bridge between the K208-E212 pair in h/bBest2 (Fig. 6e), such that Q208 points directly into the central axis of hBest1 (radius 1.8 Å), whereas its counterpart in cBest1 points away from the central axis (Fig. 6h and Supplementary Fig. 9d)[13]. This structural divergence is rooted from their different primary sequences: in cBest1, the side chain of Q208 fits in a pocket enabled by G209, whereas a serine at this position in hBest1 (S209) makes a hydrogen bond with Q208 (Supplementary Fig. 9d), causing the amino group of the Q208 side chain to point directly in towards the central axis. Thirdly, a large ion-like density is captured at the center of Q208 in 6.1% of $Ca^{2+}$-bound hBest1 particles (Fig. 6f, g and Supplementary Fig. 9b, c), which is completely absent at the cBest1 aperture (Fig. 6h)[13]. In this putative ion-bound state, I205 takes on an alternative conformation, such that the gamma-1 and delta carbons ($-CH_2CH_3$ group) point away from the central axis,

instead of towards it, while the gamma-2 carbon (-CH$_3$ group) points into the central axis (Fig. 6f). This I205 flip dilates the radius of the aperture from 0.9 Å in the closed state to 1.7 Å in the ion-bound state, potentially representing a partially open aperture.

Functionally, the critical role of I205 in hBest1 gating has been shown in our previous studies: hBest1 I205A, replacing an isoleucine with a much shorter side-chained alanine, conducted significantly bigger currents than WT hBest1 both in the absence and presence of 1 μM Ca$^{2+}$ [14,17]. To examine the functional contribution of Q208 and N212, hBest1 Q208A and N212A mutants were generated and individually transfected into HEK293 cells for patch clamp. Both of them showed similar overall and membrane expression levels compared to WT hBest1 (Supplementary Fig. 5c), and conducted robust leak currents in the absence of Ca$^{2+}$ (Fig. 6i–k), indicating deficient channel gating at the aperture. Moreover, the currents conducted by Q208A could no longer be stimulated by Ca$^{2+}$ (Fig. 6i). These results suggest a critical involvement of both Q208 and N212 in hBest1 gating, which is consistent with the channel structure.

## Discussion

In this study, we solved the human bestrophin structures in four (Ca$^{2+}$-unbound closed, Ca$^{2+}$-bound closed, Ca$^{2+}$-bound partially open, and Ca$^{2+}$-bound fully open) states. The overall architecture of hBest1 and hBest2 resembles the pentameric assembly previously shown by KpBest, cBest1 and bBest2, and retains the two landmark permeation constrictions, the neck and aperture, in the ion conducting pathway. However, the aperture is formed by I205/Q208/N212 and S205/K208/E212 in hBest1 and hBest2, respectively, in contrast to that formed by just one or two residues in KpBest, cBest1 and bBest2 [12–14]. Moreover, unlike the static apertures in previous studies, the apertures in hBest1 and hBest2 both display a distinct conformation in the presence of Ca$^{2+}$, potentially representing a partially and fully open conformation of the aperture, respectively, as dilations take place at position 205, with an ion-like density captured at position 208. Comparison of the apertures in all available bestrophin structures strongly suggests that residues 205, 208 and 212 are the three key aperture-forming components with paralog- and species-specific compositions.

Although the necks of hBest1 and hBest2 are formed by the same set of three residues (I76/F80/F84) as in cBest1 and bBest2 [12,13], we discovered a partially open conformation of this gate in Ca$^{2+}$-bound hBest1, in contrast to the fully open neck in hBest2 under the same conditions. These divergences at the neck and aperture provide the structural basis for the significantly smaller currents from hBest1 compared to hBest2 in electrophysiological recordings [12], as well as their different physiological roles. We did not find any particles with a fully open neck in hBest1 datasets, suggesting that the partially open conformation is a steady state of the neck in hBest1. Although a fully open neck was observed in hBest1$_{1–345}$, it is unclear whether or how the neck in hBest1, in the presence of the AS, adapts to the fully open conformation in physiological conditions.

Comparison of different conformations of hBest1/hBest2 revealed an important gating mechanism mediated by the AS, which wraps around the channel periphery in an inter-protomer manner and concentrically constrains the neck from opening. Our structural and functional results consistently support a dual-factor gating model: dilation of the neck requires conformational changes which are induced by the binding of Ca$^{2+}$ to the Ca$^{2+}$-sensor but sterically prohibited when the AS is bound in place. In the absence of Ca$^{2+}$, although the binding of the AS is dynamic and the constriction could be unleashed, spontaneous opening is

extremely rare without the binding of Ca$^{2+}$, consistent with the tiny currents conducted by hBest1 and hBest2 under no Ca$^{2+}$ conditions. Upon Ca$^{2+}$-binding, a cascade of conformational changes take place in AS-unbound channels, resulting in neck opening which is functionally observed as Ca$^{2+}$-dependent currents. Higher percentages of open neck particles were found in the 5 mM Ca$^{2+}$ datasets compared to the 1 μM Ca$^{2+}$ datasets and were all accompanied with the disordering of the AS, suggesting an influence of Ca$^{2+}$ on the binding dynamics of the AS. We speculate that Ca$^{2+}$ antagonizes and promotes the binding of AS to the channel periphery at low and high concentrations in steady states, respectively, correlating to channel activation and inactivation. Remarkably, removing the AS from hBest1 and hBest2 results in truncation mutants that both exhibit a fully open neck in 100% of Ca$^{2+}$-bound particles in cryo-EM and conduct indistinguishable and significantly elevated currents in patch clamp, while swapping the AS between hBest1 and hBest2 results in chimeric mutants with exchanged channel activities. Therefore, the AS determines bestrophin channel specificity among paralogs and species by serving as a tuner of the neck.

## Methods

**Cell lines**. HEK293 cells authenticated by short tandem repeat DNA profiling were kindly gifted from Dr. Henry Colecraft at Columbia University. DMEM (4.5 g L$^{-1}$ glucose, Corning 10013CV) supplemented with 10% FBS and 100 μg ml$^{-1}$ penicillin-streptomycin was used for HEK293 cell culture. No mycoplasma contamination was found by DAPI staining.

**Transfection**. 20–24 h before transfection, HEK293 cells were detached by treatment with TrypLE for 5 min at room temperature and split into new 3.5-cm culture dishes at 50% confluency. PolyJet transfection reagent (SignaGen SL100688) was used to transfect HEK293 cells with plasmids bearing the indicated WT or mutant channels (1 μg). The transfection mix was removed after 6–8 h, and cells were washed with PBS and cultured in supplemented DMEM. 24 h after transfection, cells were detached by trypsin and split onto fibronectin-coated glass coverslips for patch clamp [22].

**Electrophysiology**. Electrophysiological analyses were conducted 72–96 h after transfection. Whole-cell patch clamp recording was performed with an EPC10 patch clamp amplifier (HEKA Electronics) controlled by Patchmaster v2x90.5 (HEKA). Micropipettes were pulled and fashioned from filamented 1.5 mm thin-walled glass (WPI Instruments), and filled with internal solution containing (in mM): 130 CsCl, 1 MgCl$_2$, 10 EGTA, 2 MgATP (added fresh), 10 HEPES (pH 7.4, adjusted by CsOH). The desired no Ca$^{2+}$ and 1 μM free Ca$^{2+}$ concentrations were obtained by adding 0 and 8.7 mM CaCl$_2$, respectively (https://somapp.ucdmc.ucdavis.edu/pharmacology/bers/maxchelator/CaMgATPEGTA-TS.htm). Series resistance was typically 1.5–2.5 MΩ, with no electronic series resistance compensation. The recipe of external solution was (in mM): 140 NaCl, 5 KCl, 2 CaCl$_2$, 1 MgCl$_2$, 10 HEPES (pH 7.4, adjusted by NaOH) and 15 glucose. Solution osmolality was 290–310 mOsm/L with glucose. Traces were acquired at a repetition interval of 4 s [23]. Currents were sampled at 25 kHz and filtered at 5 or 10 kHz. I–V curves were generated from a group of step potentials (−100 to +100 mV from a holding potential of 0 mV). Experiments were conducted at room temperature (23 ± 2 °C).

**Electrophysiological data collection and analyses**. Whole-cell patch clamp data were processed off-line in Patchmaster v2x90.5. Statistical analyses were performed using built-in functions in Origin.

**Molecular cloning**. All constructs were made by site-directed mutagenesis PCR with the In-fusion Cloning Kit (Clontech) and verified by sequencing. The sequences of cloning primers are summarized in Supplementary Table 2 and contained in the Source Data file.

**Immunoblotting**. Cell pellets were extracted by the M-PER mammalian protein extraction reagent (Thermo Fisher Scientific, 78501) or Mem-PER Plus membrane protein extraction kit (Thermo Fisher Scientific, 89842) with proteinase inhibitors (Roche, 04693159001), and the protein concentration was quantified by a Bio-Rad protein reader. After denaturing at 95 °C for 5 min, the samples (20 μg) were run on 4–15% gradient SDS-PAGE gel at room temperature, and wet transferred onto nitrocellulose membrane at 4 °C. The membranes were incubated in blocking buffer containing 5% (w/v) non-fat milk for 1 h at room temperature, and subsequently incubated overnight at 4 °C in blocking buffer supplemented with primary antibodies: rabbit anti-GFP (1:1,000 Fisher Scientific, A6455), mouse anti-β-actin

(1:2,000 Invitrogen, MA5-15739). Fluorophore-conjugated secondary antibodies IRDye® 680RD goat anti-rabbit IgG (LI-COR, 926–68071) and IRDye® 680RD goat anti-mouse IgG (LI-COR, 926–68070) were used at a concentration of 1:10,000 and an incubation time of 1 h at room temperature, followed by infrared imaging.

**Protein production and purification.** hBest1 and hBest2 were expressed and purified as follows[24]. All constructs were expressed by Bacmam system in HEK293-F cells at 30 °C for 3 to 4 days. Expression was validated by the presence of GFP on the membrane on the third or fourth day. Cells were pelleted and washed with buffer or a small volume of media prior to freezing at −80 °C. Cell pellets were resuspended in buffer containing 50 mM HEPES (pH 7.8), 300 mM NaCl, 5% glycerol, 20 mM imidazole, 1 mM PMSF, and EDTA-free protease inhibitor cocktail (Sigma S8830) and lysed by sonication on ice. Crude insoluble material was removed by low-speed centrifugation (~7,000 × g) for 10 min. Membranes were isolated from the supernatant by ultracentrifugation at >100,000 × g for 1 h. The gelatinous crude membranes were collected and stored or directly used for protein production.

Membranes were resuspended in the same resuspension buffer at 40–100 mg/mL and extracted with 1.5 % (w/v) GDN for 1–2 h on a gentle over-under rotator at 4 °C. Insoluble material was removed by centrifugation at 15,000 × g for 30 min and the detergent-extracted solution was incubated with Ni-NTA resin for 30–60 min. The lysate was allowed to flow over the resin multiple times and more resin was added if the flow through remained GFP-positive. The resin was washed with 5 column volumes (CV) of wash buffer containing 75 mM imidazole, 25 mM HEPES (pH 7.8), 500 mM NaCl, 5% glycerol, and 0.005% GDN, followed by a final wash with the same buffer containing 125 mM imidazole. Protein was eluted in elution buffer containing 500 mM imidazole, 25 mM HEPES (pH 7.8), 200 mM NaCl, 5% glycerol. The eluted protein solution (~10–15 mL per 1–1.5 L of starting culture) was concentrated with a 100 kDa molecular weight cutoff (MWCO) filter concentrator and buffer exchanged multiple times with gel filtration buffer containing 5% glycerol until the imidazole concentration was ~25 mM. The protein was concentrated to ~1–3 mg/mL for overnight treatment with TEV protease to remove the GFP-HIS tag. On the next morning, cleaved GFP-HIS and TEV-HIS were removed with fresh Ni-NTA resin. bBest2 and hBest2 tend to stick to the resin and required multiple washes with buffer containing 75 mM imidazole and 500 mM NaCl to recover the protein. hBest1 tended not to stick to the Ni-NTA resin, but the resin was always washed thoroughly to recover bound protein. The flow through was concentrated and if GFP persisted, more Ni-NTA resin was added until the solution appeared clear. The protein solution was concentrated to 0.5 mL and injected onto a Superose 200 increase 10/300 gel filtration column equilibrated with gel filtration buffer containing 200 mM NaCl, 40 mM HEPES, pH 7.8, and 0.005% (w/v) GDN (glycerol was always omitted at this stage). The protein eluting at ~10 mL was concentrated to 3–5 mg/mL with a Sartorius Vivaspin 500 (100 kDa MWCO) and used for cryo-EM studies.

**Cryo-EM grid production and data collection.** All grids were prepared on a FEI Vitrobot Mark IV. All cryo-EM grids were made with UltrAuFoil R0.6/1 or R1.2/1.3 300 mesh (Quantifoil) and were plasma treated in batches of up to 4 immediately before use. The Vitrobot was equilibrated to 100% humidity and 10 °C for at least 1 h. prior to use. For all conditions using EGTA, the EGTA was incubated with the protein for at least 1 h prior to grid production and all Ca²⁺- EGTA solutions were prepared according to the MaxChelator program. All EGTA conditions were prepared with ashless Ca²⁺-free filter paper (Whatman #40), while Ca²⁺-containing conditions were blotted with Whatman #1.

All Krios data collection and screening was performed at the Columbia cryo-EM Core Facility with data collection and targeting performed through Leginon[25]. Ice thickness was monitored using inelastic scattering function as implemented in Appion and cryoSPARC Live was used for real-time imaging diagnostics[26,27]. Final grids were imaged on Columbia's Krios G3i equipped with K3 direct electron detector and BioQuantum energy filter at a nominal magnification of 105,000× mag corresponding to a physical pixel size of 0.83 Å²/pixel. Movies collected without CDS mode contained 50 frames collected over 2.5 s. with a dose rate of 16 e⁻/pix/s for a total dosage of 58 e⁻/Å² with 1.16 e⁻/Å²/frame. Those collected in CDS mode contained 50 frames collected over 5 s. with a dose rate of 8 e⁻/pix/s. All movies were collected with an energy filter slit width of 20 eV.

**Cryo-EM image processing.** The same general image processing pathway was performed for all datasets. Movies underwent motion correction by MotionCor2 through the Relion 3.0 GUI with a 7 × 5 patch, which allowed Bayesian polishing at a later stage[22,28]. Motion-corrected micrographs were imported to cryoSPARC for subsequent processing[23]. Micrographs underwent CTF estimation by Patch CTF, followed by template-based picking using low-pass filtered 2D classes from one of the previously processed bestrophin datasets. Individual particle images from the template pick job were extracted with binning to 4–6 Å/pix for subsequent 2D classification in cryoSPARC. Multiple rounds of 2D classification were used to remove particles not corresponding to bestrophin particles, which may be debris, gold substrate, or damaged protein. Particles were then extracted

with binning to 1-2 Å²/pix for 3D classification in Relion 3.0 after csparc2star with pyem[29].

The fully open neck of hBest2, partially open neck of hBest1, and the hBest1 class with ion bound at Q208 were obtained by the same general focused 3D classification strategy in Relion 3.0 with a tight mask on the region of interest and skipping alignments and shifts (Supplementary Figs. 1, 2). The partially open neck of hBest1 was identified by 3D classification of 455k particles into 8 classes with the indicated settings in Supplementary Fig. 1c. The open state comprising 5.7% of total particles was subjected to heterogeneous refinement into two classes and the best class was used for model building after a single round of homogeneous refinement with C5 symmetry. The same particle set was subjected to homogeneous refinement with C1 symmetry to verify that the central density is not an artifact of imposed symmetry (Supplementary Fig. 1d). The class with density at Q208 was obtained by subjecting the same particle set to 3D classification with a mask on the aperture region, revealing a 6.1% class with apparent density at Q208. This class was subjected to homogeneous refinement in C5 symmetry and C1 symmetry (Supplementary Fig. 1e). The hBest2 class with fully open neck was identified by 3D classification with the indicated settings in Supplementary Fig. 2d. hBest2 datasets were also subjected to 3D classification with a mask on the aperture and no deviation from the consensus refinement was identified.

All final particle sets were subjected to homogeneous refinement or non-uniform refinement with C5 symmetry unless indicated otherwise[30]. In most cases, on-the-fly per-particle defocus optimization and per-group CTF parameter optimization including higher order aberration correction were utilized during refinement. Bayesian polishing was performed on final particle sets for most datasets, and Ewald curvature correction was tested on the best refinements with no benefit. All reported resolutions are based on the FSC curves calculated from two independently refined half maps with the gold standard cutoff of 0.143, as implemented in cryoSPARC.

Final maps used for model building and refinement were obtained by density modification in phenix (phenix.resolve), Servalcat (diffmap_fo.mrc), autosharpening in phenix or by sharpening to a b-factor determined by Guinier plot[31–33]. The optimal map was determined subjectively by visual inspection of high-resolution and low-resolution details to maintain optimal interpretation of regions of interest. The highest resolution hBest2 map extended to 1.78 Å and that for hBest1 to 1.82 Å, or 93% and 91% of Nyquist for a pixel size of 0.83 Å²/pix, respectively. Model building and map interpretation were aided by resampling these near-Nyquist maps on a finer grid using cryoEM tools in Coot 0.9.6.2-pre (Sharpen/Blur tool, resample factor 1.5 to 2.0). Lipids were often oversharpened and their placement was aided by building into a blurred or undersharpened map which better resolved these low-resolution features.

**Model building, refinement, and validation.** Initial models for hBest1 and hBest2 were generated with SWISS-MODEL based on the bBest2 (PDB 6VX7) template[34]. The PDB was rigid body fit into the cryo-EM map and subjected to multiple iterations of refinement in coot and phenix real space refinement, and/or REFMAC5 (Servalcat)[35–38]. Validation was performed with comprehensive cryo-EM validation tools in phenix, including MolProbity, while density fit was assessed with EMRinger and difference maps produced by Servalcat were used to assess map-model differences[32,39,40].

**3Dvariability analysis.** 3D-variability analysis (3DVA) was used to visualize movements and the relative order and disorder in the various datasets[41]. For the hBest2 dataset, 986,365 particles underwent homogeneous refinement (C5 symmetry enforced), followed by symmetry expansion and then 3DVA was performed with a tight mask on the density corresponding to protein, while excluding the majority of the micelle. 3DVA was run with 2 modes, a filter resolution of 3.5 Å, and a high-pass filter of 20 Å. The hBest1 dataset was analyzed in the same manner with 454,564 particles.

To generate movies, 3DVA was visualized in "simple" mode with 20 frames and with a rolling window of 0 frames (setting: "Intermediates: window (frames) = 0") and visualized as a volume series in Chimera[42]. Cluster mode was used to separate particles along the component of interest for analysis of AS binding in Fig. 3. Particles were clustered and subjected to local refinement with a global mask in C1 symmetry to generate the maps corresponding to the first and last frames of 3DVA. These maps were then cropped to the same physical box size and visualized at sigma 4 to allow standardized visualization of the relative order of the regions of the AS.

**Statistics and reproducibility.** A sufficient number of samples were examined to reach statistical conclusion according to the specific method utilized in that experiment. Statistically significant differences between means ($P < 0.05$) were determined using Student's $t$ test for comparisons between two groups, or using one-way ANOVA followed by Bonferroni *post hoc* analyses for comparisons involving more than two groups. Data are presented as mean values ± SEM.

**Reporting summary**. Further information on research design is available in the Nature Research Reporting Summary linked to this article.

## Data availability

The data that support this study are available from the corresponding authors upon reasonable request. The cryo-EM density maps have been deposited in the Electron Microscopy Data Bank (EMDB) under accession numbers EMD-27131 (hBEST1 1μM closed), EMD-27132 (hBEST1 5 mM closed), EMD-27133 (hBEST1 1μM neck partially open), EMD-27134 (hBEST1 1μM aperture partially open), EMD-27135 (hBEST1 EGTA closed), EMD-27137 (hBEST1 345 open), EMD-27127 (hBEST2 1μM closed), EMD-27128 (hBEST2 5 mM closed), EMD-27129 (hBEST2 open), EMD-27130 (hBEST2 EGTA closed), and EMD-27136 (bBEST2 345 open). The coordinates have been in the RCSB Protein Data Bank (PDB) under accession codes 8D1I [https://doi.org/10.2210/pdb8D1I/pdb] (hBEST1 1μM closed), 8D1J [https://doi.org/10.2210/pdb8D1J/pdb] (hBEST1 5 mM closed), 8D1K [https://doi.org/10.2210/pdb8D1K/pdb] (hBEST1 1μM neck partially open), 8D1L [https://doi.org/10.2210/pdb8D1L/pdb] (hBEST1 1μM aperture partially open), 8D1M [https://doi.org/10.2210/pdb8D1M/pdb] (hBEST1 EGTA closed), 8D1O [https://doi.org/10.2210/pdb8D1O/pdb] (hBEST1 345 open), 8D1E [https://doi.org/10.2210/pdb8D1E/pdb] (hBEST2 1μM closed), 8D1F [https://doi.org/10.2210/pdb8D1F/pdb] (hBEST2 5 mM closed), 8D1G [https://doi.org/10.2210/pdb8D1G/pdb] (hBEST2 open), 8D1H [https://doi.org/10.2210/pdb8D1H/pdb] (hBEST2 EGTA closed), and 8D1N [https://doi.org/10.2210/pdb8D1N/pdb] (bBEST2 345 open). Source data for Figs. 5 and 6, and Supplementary Figs. 6 and 8 are provided with this paper in the Source Data file. Source data are provided with this paper.

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

## Acknowledgements

We thank the Center on Membrane Protein Production and Analysis (New York Structural Biology Center, supported by NIH GM116799) and Robert Grassucci, Zhening Zheng, Jing Wang, Chi Wang, and Ying-Chih Chi of Columbia Cryo-EM Core Facility for assistance with Krios operation and screening. We thank Zhen Gong and Jing Wang for providing cells for protein production. We also thank Oliver Clarke for helpful discussions on cryo-EM processing and model building and Harry Kao for providing access to cryoSPARC Live during data collection. All cryo-EM data were collected at Columbia Cryo-EM Core Facility. A.P.O. was supported by NIH grant EY030763, W.A.H. was supported by NIH grant GM107462, and T.Y. was supported by NIH grants (GM127652, EY028758), the Irma T. Hirschl/Monique Weill-Caulier Research Award (CU20-4313) and Schaefer Research Award.

## Author contributions

A.P.O. designed research, performed protein purification and cryo-EM experiments, analyzed data, made figures and helped with writing the paper; J.W. designed research, performed patch clamp recordings, analyzed data and made figures; A.K. designed research and performed molecular experiments; Z.C. made constructs; W.A.H. designed research and analyzed data; Y.Z. and T.Y. designed research, analyzed data, made figures and wrote the paper.

## Competing interests

The authors declare no competing interests.
