## [Peer Review File · Nature Communications]

Structures and gating mechanisms of human bestrophin anion channelsReviewers' Comments:

Reviewer #1:

Remarks to the Author:

Owji et al. use cryo-EM to determine the structure of human Bestrophin-1 and Bestrophin-2. The authors determine the structure in absence and the presence of two different concentrations of calcium. For the first time, they are able to obtain a subset of structures showing the open configuration of the channel. Interestingly, these structure shows both a dilation of the so-called "neck" and the opening of the "aperture". These structures also pointed out to a new possible molecular mechanisms controlling conduction in bestrophin channels. Indeed, the authors proposes that the C terminus of the protein works as "auto-inhibitory segment" controlling the ion conductance. These data are novel and interesting , but I think some control experiments are need to fully establish the conclusions of the manuscripts. Here my major concerns:

- The authors reports here and in one of their previous work that hBest1 generates significantly smaller current than hBest2 when heterologous expressed and here they found possible structural mechanism for this differences. However, previous data show no difference in amplitude of hBest1 and hBest2 current (Sun et al., 2002, PMID 11904445). An obvious explanation of the difference in hBest1/hBest2 current amplitude is the difference in membrane expression of the two proteins. The authors should check if the membrane trafficking could explain the observed difference.
- To investigate the role of AS, the authors determined the structure of 1-345 portion of hBest1 and bBest2 founding a that 100% of particles show a fully open neck. What about the "aperture"? Are the channels really open? Moreover, what about structure of these truncated channels in the absence of calcium? Does the removal of AS completely abolish the calcium-dependent gating?
- Line 144: line 114: Do the proportion of closed/open channels in the presence of the ligand relate to ligand efficacy? Are there other reason for a conformational shift?
- The statical analysis of patch clamp data in Figure S5.1 should not be performed using an unpaired t-test. An ANOVA (but see later) followed by multicomparison test should be used. The normality of the patch clamp data should be test using statistical test before using a parametric test.

Some minor points:

Line 499: Caption title is misleading, as it emphasize the closed state while the figure is also about the putative open and partially open states

Line 289-290: This sentence is misleading. hBest1 is also permeable to bicarbonate even if less than hBest2 (please see Qu et al., 2008 PMID 18400985). Moreover mBest1 seems to be permeable to big molecules such as GABA (Lee et al., 2010, PMID 20929730).

Line 376: Link to maxchelator is not working. The composition of the solution used should be added to help the reproducibility of the data

Line 532-552: In both legend is reported 1.2 μ M of calcium while in the text 1 μ M. What is correct?

line 31: "unified activities" sounds awkward and hard to understand

Perhaps a final figure with a cartoon summarizing the proposed gating model might be useful.

Reviewer #2:

Remarks to the Author:

The authors report cryo-EM structures of bestrophin 1 and 2 channels in multiple conformations including closed, partially open, and open states. In addition, the authors structurally and functionally identified a C-terminal auto-inhibitory segment that is critical for regulation of channel activity because chimera constructs resulted in altered channel properties. The structures appear to be of high quality, with resolutions up to 1.9 angstroms. Overall, this is an interesting study based on solid data. The following points need to be addressed before publication.

One of the major novel findings of this manuscript is the C-terminal auto-inhibitory segment, which appeared to disengage from the channel core during activation. Figure 3 showed density maps to

make this point. However, to provide a more rigorous comparison, these cryo-EM maps need to be filtered at the same resolution and compared at the same sigma levels.

Fig. 3 map densities are of poor quality and improvement could be made. Panel e and g: 'partially open' and 'fully open' seem to be mislabeled.

Page 7 describes C-terminal AS segment interaction with the core. Specific residues or interactions were discussed in the main text, but Figures did not clearly show these points. For instance, "H156 and A160 of helix S2f, F151 of helix S2e, and P346 and T348 of the next protomer over". It is thus difficult to follow the statement in the main text. Also, R357 is mentioned in the main text but Q357 shown in Figure 3d.

In Figure 5 and 6, the authors used steady-state current densities for truncations, chimeras, and point mutations to assess channel properties, but did not show that these constructs had similar surface expression levels (such as verification by Western Blot). This information would be critical for valid comparison of these mutant channels.

In Figure 1, the title states "closed state structures", but partially open and open structures are shown in Panel e and f.

Reviewer #3:

Remarks to the Author:

This paper by Owji et al. aims to understand the mechanisms of gating of bestrophin calcium-activated chloride channels. Bestrophins are less well understood at the molecular level than members of the TMEM16 family, which are also calcium-activated chloride channels, but they are crucial in cell physiology of several organs including eye, colon, kidney, and pancreas. They have been linked to several eye diseases, including glaucoma and a type of macular degeneration. The present study makes a major contribution to understanding how bestrophin channels are gated open by calcium. Previous cryo-EM and X-ray diffraction studies used truncated and not full-length proteins and did not detect open state structures. Further, this is the first study to examine human human BEST proteins. This study presents novel bestrophin structures in closed, partially open, and fully open states. These structures provide very valuable insights into gating mechanisms of these channels. These structural insights are then tested by mutagenesis and patch clamp. Further, the comparison of BEST1 and BEST2 structures reveals insights into the functions of key amino acids and differences in the two proteins that were not previously appreciated. Overall, the work is carefully done and the data support the conclusions.

Major comment.

The electrophysiological results are difficult to interpret without information about relative expression levels. Western blots should be shown to evaluate expression levels of various constructs. Furthermore, what is the evidence that these currents are actually mediated by bestrophins? This is especially important for the large hBest1/2(1-345) currents in the absence of Ca²⁺ – how do you know this is not some non-specific leak? What is the ionic selectivity of these currents? Can the authors use pharmacological tools to rule out leaks? Also, the authors should show representative current traces (probably in supplementary figures).

Minor comments.

1. Figure 2. What software was used to determine the pore density? Please provide a quantitative color key and, in addition to these images, please provide plots of pore diameter vs. distance along the pore.

2. Figure 3a. Figure 3 was hard to decipher. (1) The linear representation of the AS and the description in the legend is confusing for several reasons. The colored segments are not to scale (for example, the 1-345 core is shorter than 345-379). Different line thicknesses or something could be used to differentiate the core from the AS. (2) Why does the linear representation read left-to-right 379 to 1, rather than right-to-left 1 to 379? (3) The legend says "the specific segments of the AS with their protomer colored copper." The wording is weird. The AS is part of the protomer. (4) In the 3D structures in a and b, there is a purple segment that is not shown in the linear representation. (5) The legend (c) should read something like "Closeup of ACR2 of protomer -1 interacting with ACR1 of protomer 0" (6) The wording of legend (d) is hard to understand.

3. In several places (for example line 145), the authors use the phrase "the next protomer over". I think it would be better to use the protomer numbers.

4. Figure 3a defines the residue numbers of ACR1, anchor, and ACR2, but the text lines 148-154 refers to residue numbers. It might be easier to read if the acronyms were used along with - or instead of - the residue numbers. In addition, lines 154-158 are long and confusing. I would suggest saying "the anchor holds the C-terminal segment to the neighboring protomer, while the ACRs bind and unbind reversibly to regulate channel gating" And then elaborate in another sentence.

5. In Fig. S3.1 legend, it would help to note again the difference in numbering of Best1 and Best2.

6. The authors should explain the 3DVA analysis in Methods.

7. Line 210. The authors do not show single channel recordings here, so should they should rephrase "channel activity" to read "currents".

8. Line 272. Please add "full-length" to the statement that these are the first structures from WT proteins. Previous structures were WT, but truncated.

We thank the reviewers for their supportive comments and constructive critiques on our work. We have prepared a revised version of the manuscript that incorporates changes to address the reviewers' questions and comments. We copy the referee comments below verbatim and respond to each of the points, describing how we have modified the paper to address their concerns.

Reviewer #1

1. *"The authors reports here and in one of their previous work that hBest1 generates significantly smaller current than hBest2 when heterologous expressed and here they found possible structural mechanism for this differences. However, previous data show no difference in amplitude of hBest1 and hBest2 current (Sun et al., 2002, PMID 11904445). An obvious explanation of the difference in hBest1/hBest2 current amplitude is the difference in membrane expression of the two proteins. The authors should check if the membrane trafficking could explain the observed difference."*

Response: As the reviewers suggested, we have added Western blot results in Figure S5.1 showing no significant difference between hBest1 and hBest2 in their membrane expression levels.

2. *"To investigate the role of AS, the authors determined the structure of 1-345 portion of hBest1 and bBest2 founding that 100% of particles show a fully open neck. What about the "aperture"? Are the channels really open? Moreover, what about structure of these truncated channels in the absence of calcium? Does the removal of AS completely abolish the calcium-dependent gating?"*

Response: The local resolution at the aperture region of 1-345 is not high enough for further classification to distinguish different states. The 1-345 truncations of hBest1 and hBest2 both conduct significantly elevated currents compared to the respective FL channels, but we do not know whether it solely reflects more opening of the neck, or more opening also takes place at the aperture.

We have tried multiple times to obtain the Ca²⁺-unbound structures of the 1-345 truncations, but it was very difficult to remove Ca²⁺ from these constructs and we always obtained a mixture of Ca²⁺-bound and -unbound particles with a mixture of the open and closed conformations at the neck. Although we were unable to separate different classes in the cryo-EM particles, our patch clamp results show that the 1-345 truncation mutants conduct Ca²⁺-independent leak currents in the absence of Ca²⁺ and can be significantly stimulated by Ca²⁺. These functional data suggest that the removal of AS promotes spontaneous channel opening but does not completely abolish the Ca²⁺-dependent gating.

3. *"Line 114: Do the proportion of closed/open channels in the presence of the ligand relate to ligand efficacy? Are there other reason for a conformational shift?"*

Response: 100% of hBest1/hBest2 particles from both 1 μ M and 5 mM Ca²⁺ conditions are bound with Ca²⁺, while a higher percentage of particles in the partially/fully open conformation was detected under the 5 mM Ca²⁺ condition, suggesting that the different closed/open ratios are not due to ligand efficacy. Our results suggest that AS unbinding is necessary for neck opening in the presence of Ca²⁺, as all the partially/fully open particles are associated with AS disordering.

4. *"The statistical analysis of patch clamp data in Figure S5.1 should not be performed using an unpaired t-test. An ANOVA (but see later) followed by multicomparison test should be used. The normality of the patch clamp data should be tested using statistical test before using a parametric test."*

Response: We have re-analyzed the data in Fig. S5.4 (original Fig. S5.1) as the reviewer suggested.

5. *"Line 499: Caption title is misleading, as it emphasize the closed state while the figure is also about the putative open and partially open states."*

Response: As the reviewer suggested, we have revised the figure title to "Cryo-EM structures of hBest1 and hBest2 in different states".

6. *"Line 289-290: This sentence is misleading. hBest1 is also permeable to bicarbonate even if less than hBest2"*

(please see Qu et al., 2008 PMID 18400985). Moreover mBest1 seems to be permeable to big molecules such as GABA (Lee et al., 2010, PMID 20929730)."

Response: We thank the reviewer for pointing it out, and have deleted this sentence to avoid misleading.

7. "Line 376: Link to maxchelator is not working. The composition of the solution used should be added to help the reproducibility of the data."

Response: We have updated the link and added the composition of the solution as the reviewer suggested.

8. "Line 532-552: In both legend is reported 1.2 uM of calcium while in the text 1 uM. What is correct?"

Response: 1 μ M is correct. We have corrected the figure legend.

9. "line 31: "unified activities" sounds awkward and hard to understand. Perhaps a final figure with a cartoon summarizing the proposed gating model might be useful."

Response: We have revised "unified" to "similar" and added a cartoon (Fig. S5.3) summarizing the model.

Reviewer #2

1. "One of the major novel findings of this manuscript is the C-terminal auto-inhibitory segment, which appeared to disengage from the channel core during activation. Figure 3 showed density maps to make this point. However, to provide a more rigorous comparison, these cryo-EM maps need to be filtered at the same resolution and compared at the same sigma levels. Fig. 3 map densities are of poor quality and improvement could be made. Panel e and g: 'partially open' and 'fully open' seem to be mislabeled."

Response: We have revised Fig. 3 using the same filter resolution and the same sigma levels for comparison, improved the quality of map densities, and corrected panel e and g labels.

2. "Page 7 describes C-terminal AS segment interaction with the core. Specific residues or interactions were discussed in the main text, but Figures did not clearly show these points. For instance, "H156 and A160 of helix S2f, F151 of helix S2e, and P346 and T348 of the next protomer over". It is thus difficult to follow the statement in the main text. Also, R357 is mentioned in the main text but Q357 shown in Figure 3d."

Response: As the reviewer suggested, we have modified Fig. S3.1 to better illustrate the specific residues\interactions described in the text. Q357 is correct in hBest2, and we have corrected the typo.

3. "In Figure 5 and 6, the authors used steady-state current densities for truncations, chimeras, and point mutations to assess channel properties, but did not show that these constructs had similar surface expression levels (such as verification by Western Blot). This information would be critical for valid comparison of these mutant channels."

Response: As the reviewer suggested, we have added Western blot results showing similar surface expression levels of the tested hBest1 and hBest2 constructs in Fig. S5.1.

4. "In Figure 1, the title states "closed state structures", but partially open and open structures are shown in Panel e and f."

Response: As the reviewer suggested, we have revised the title to "Cryo-EM structures of hBest1 and hBest2 in different states".

Reviewer #3

1. "The electrophysiological results are difficult to interpret without information about relative expression levels. Western blots should be shown to evaluate expression levels of various constructs. Furthermore, what is the

evidence that these currents are actually mediated by bestrophins? This is especially important for the large hBest1/2(1-345) currents in the absence of Ca – how do you know this is not some non-specific leak? What is the ionic selectivity of these currents? Can the authors use pharmacological tools to rule out leaks? Also, the authors should show representative current traces (probably in supplementary figures)."

Response: As the reviewer suggested, we have added Western blot results showing similar surface expression levels of the tested constructs in Fig. S5.1 and added representative current traces in Fig. S5.2. In regard to the specificity of bestrophin-mediated currents, cells transiently expressing the WT and 1-345 conducted currents that are decreased to ~ 30-40% upon the treatment of Cl⁻ channel blocker NFA (Fig. S5.2), while untransfected cells only exhibited tiny currents (< 5 pA/pF, Fig. 6k). We are actively working on the ionic selectivity of the bestrophin channels, and respectfully think it would be suited for a separate report as this manuscript is focused on gating.

2. *"Figure 2. What software was used to determine the pore density? Please provide a quantitative color key and, in addition to these images, please provide plots of pore diameter vs. distance along the pore."*

Response: HOLE was used to compute the pore dimensions, while Chimera was used to visualize pore densities for figures (We have added this information in Fig. 1 legend). As the reviewer suggested, we have added a quantitative color key and plots of pore radius vs. distance along the pore in Figs. 1 and 2.

3. *"Figure 3 was hard to decipher. (1) The linear representation of the AS and the description in the legend is confusing for several reasons. The colored segments are not to scale (for example, the 1-345 core is shorter than 345-379). Different line thicknesses or something could be used to differentiate the core from the AS. (2) Why does the linear representation read left-to-right 379 to 1, rather than right-to-left 1 to 379? (3) The legend says "the specific segments of the AS with their protomer colored copper." The wording is weird. The AS is part of the protomer. (4) In the 3D structures in a and b, there is a purple segment that is not shown in the linear representation. (5) The legend (c) should read something like "Closeup of ACR2 of protomer -1 interacting with ACR1 of protomer 0" (6) The wording of legend (d) is hard to understand."*

Response: We have revised Figure 3 and the legend according to the reviewer suggestions. (1) Distinct thickness and length are used for the core; (2) The linear representation is horizontally flipped; (3) "...with their protomer..." is revised to "...and the channel core of protomer 0..."; (4) The purple segment is re-colored copper as it is part of the core; (5) The legend (c) is revised as suggested; (6) The legend (d) is re-written.

4. *"In several places (for example line 145), the authors use the phrase "the next protomer over". I think it would be better to use the protomer numbers."*

Response: We have added the protomer numbers as the reviewer suggested.

5. *"Figure 3a defines the residue numbers of ACR1, anchor, and ACR2, but the text lines 148-154 refers to residue numbers. It might be easier to read if the acronyms were used along with - or instead of - the residue numbers. In addition, lines 154-158 are long and confusing. I would suggest saying "the anchor holds the C-terminal segment to the neighboring protomer, while the ACRs bind and unbind reversibly to regulate channel gating" And then elaborate in another sentence."*

Response: The reviewer's comment on the acronyms is very well taken. As the conformational changes in these residues were described for the first time in (original) lines 148-154, we respectfully think that it seems more seamless to assign acronyms (in original lines 168-172) after a full description of them in this paragraph. We have revised lines 154-158 as the reviewer suggested.

6. *"In Fig. S3.1 legend, it would help to note again the difference in numbering of Best1 and Best2."*

Response: Added.

7. *"The authors should explain the 3DVA analysis in Methods."*

Response: As the reviewer suggested, we have added explanation for the 3DVA analysis in the Methods.

8. *“Line 210. The authors do not show single channel recordings here, so should they should rephrase “channel activity” to read “currents”.”*

Response: Revised.

9. *“Line 272. Please add “full-length” to the statement that these are the first structures from WT proteins. Previous structures were WT, but truncated.”*

Response: Added.

Reviewers' Comments:

Reviewer #1:

Remarks to the Author:

The authors addressed all the points I raised. I strongly suggest to publish the paper in the revised version.

Reviewer #2:

Remarks to the Author:

The authors have addressed my concerns and the manuscript is ready for publication. I would like to congratulate the authors on this important work.

Reviewer #3:

Remarks to the Author:

The authors have answered most of my concerns.

In Fig. 5S1, the "membrane" fraction appears to be crude fraction that sediments between 7,000 g - 100,000 g. While this fraction contains membranes, I think it is a stretch to call it a membrane fraction. I would prefer "particulate fraction", but if the authors insist, I would accept "crude membrane fraction".